Global mental health delivery; healthcare workers; task sharing; low-income countries; peer volunteers

**Corresponding author:**
Miya L. Barnett;
Email: mbarnett@ucsb.edu

# Effective training practices for non-specialist providers to promote high-quality mental health intervention delivery: A narrative review with four case studies from Kenya, Ethiopia, and the United States

Miya L. Barnett[1] ⓘ, Eve S. Puffer[2] ⓘ, Lauren C. Ng[3] ⓘ and Florence Jaguga[4]

[1]Department of Counseling, Clinical, and School Psychology, Gevirtz Graduate School of Education, University of California, Santa Barbara, Santa Barbara, CA, USA; [2]Department of Psychology and Neuroscience, Global Health Institute, Duke University, Durham, NC, USA; [3]Department of Psychology, University of California, Los Angeles, Los Angeles, CA, USA and [4]Department of Mental Health, Moi Teaching and Referral Hospital, Eldoret, Kenya

## Abstract

Mental health needs and disparities are widespread and have been exacerbated by the COVID-19 pandemic, with the greatest burden being on marginalized individuals worldwide. The World Health Organization developed the Mental Health Gap Action Programme to address growing global mental health needs by promoting task sharing in the delivery of psychosocial and psychological interventions. However, little is known about the training needed for non-specialists to deliver these interventions with high levels of competence and fidelity. This article provides a brief conceptual overview of the evidence concerning the training of non-specialists carrying out task-sharing psychosocial and psychological interventions while utilizing illustrative case studies from Kenya, Ethiopia, and the United States to highlight findings from the literature. In this article, the authors discuss the importance of tailoring training to the skills and needs of the non-specialist providers and their roles in the delivery of an intervention. This narrative review with four case studies advocates for training that recognizes the expertise that non-specialist providers bring to intervention delivery, including how they promote culturally responsive care within their communities.

## Impact statement

Non-specialist providers are a critical workforce to address the global mental health burden, especially in the wake of COVID-19. However, questions remain regarding the best ways to train non-specialists to deliver high-quality psychological and psychosocial interventions. This article poses four questions that need to be considered when training non-specialists: 1) who the providers are, 2) what roles they have, 3) which skills are needed, and 4) how initial and ongoing training are conducted. Case studies and a brief narrative review of the literature on training non-specialists provide considerations to tailor training efforts to enhance non-specialist delivery of interventions.

## Introduction

The substantial worldwide gap between individuals who need mental health services, and the provision of these services has been well recognized for decades. Prior to the COVID-19 pandemic, it was estimated that at least 30% of the population worldwide experienced a mental disorder (Vigo et al., 2016), with 72–93% of individuals in low- and middle-income countries (LMICs) not receiving needed care (Roll et al., 2013; World Health Organization, 2019). Additionally, marginalized populations in high income countries (HICs), including immigrants, refugees, and racial and ethnic minoritized populations, faced similar gaps in the receipt of mental health services (Dua et al., 2011; Barnett et al., 2018*b*). Mental health needs and barriers to receiving care have only been exacerbated by the COVID-19 pandemic, with individuals impacted by death of family members and friends, social distancing, and job losses (Kola, 2020). Indeed, the COVID-19 pandemic has highlighted the need to "abandon the HIC versus LMIC dichotomy," and focus on bidirectional learnings across settings, as public health responses in some LMICs has led to minimal loss of lives, whereas HICs, such as the United States have had high mortality rates (Kola et al., 2021, p. 542).

To address global mental health needs, the World Health Organization (WHO) developed the Mental Health Gap Action Programme (mhGAP), which provided guidance on increasing access

to care through task sharing the delivery of mental health services with non-specialist providers within primary health care and community settings (World Health Organization, 2008). According to the mhGAP, non-specialists can provide psychosocial interventions or psychological interventions. Psychosocial interventions include providing psychoeducation on mental health conditions and appropriate treatments, teaching stress management strategies, and supporting functioning in daily activities. Psychological interventions include the delivery of evidence-based practices, such as behavioral activation, cognitive behavioral therapy, or family counseling. Accumulating evidence has shown that training non-specialist providers to deliver evidence-based psychological interventions is a promising strategy to help address disparities in access and quality of care (van Ginneken et al., 2013; Singla et al., 2017; Barnett et al., 2018a). As mental health needs expand in response to COVID-19 and other humanitarian crises, the need to scale-up task-sharing models across the globe has been increasingly recognized (Kola, 2020). Even with abundant evidence that non-specialists can enhance access to mental health services, questions remain globally about the training that non-specialists need to successfully scale-up and sustain effective task-sharing models (Betancourt and Chambers, 2016; Bunn et al., 2021).

This article will provide a brief narrative overview of the literature related to the initial (e.g., workshops) and ongoing (e.g., consultation and supervision) training needed for non-specialist providers to deliver psychosocial and psychological interventions with fidelity to address the global mental health gap. The narrative review builds on systematic reviews of non-specialist delivery of psychosocial and psychological interventions that identified descriptions of training and supervision within the research literature (Singla et al., 2017; Barnett et al., 2018b), along with a systematic review specific to mental health training courses for non-specialist providers (Caulfield et al., 2019). Notably, all three of these systematic reviews identified wide variety in processes for training and supervising non-specialist providers, including the length of training, the content covered, and the outcomes measured. Further clarity on effective training for non-specialists was noted as critical for the success of task-sharing interventions. Based on previous recognition that majority of research on non-specialists providing psychosocial and psychological interventions provides limit details regarding training and supervision (Singla et al., 2017; Barnett et al., 2018b; Caulfield et al., 2019) this article includes illustrative case studies to highlight the importance of recognizing there is not a "one size fits all" approach to training non-specialist providers. Case studies are especially helpful to understand the *how* and *why* of phenomenon in-depth (Schoch, 2020). Specifically, four questions organize considerations regarding how to tailor training for non-specialists: 1) who the providers are, 2) what roles they have, 3) which skills are needed, and finally, 4) how initial and ongoing training are conducted. Additionally, this article advocates for an approach that seeks to train a workforce equipped to address the global mental health gap, as opposed to an approach that solely focuses on implementing individual interventions.

## Method

### Design

This article used an explanatory multiple case study design to illuminate shared and unique processes present in training and supervising non-specialists across different countries and contexts (Greene and David, 1984; Schoch, 2020). This approach incorporates four main features of multiple case study designs: 1) a conceptual framework of roles that non-specialists need training for to deliver psychosocial and psychological interventions (Barnett et al., 2018b), 2) a sampling plan to highlight a breadth of non-specialist examples, 3) procedures for collecting data about each individual case study, and 4) a cross-case study analysis using qualitative synthesis procedures (Greene and David, 1984). Purposeful case sampling was used to identify information rich cases, with a recognition that small sample sizes (3–4 case studies) can help to understand a phenomenon in-depth (Schoch, 2020). Four case studies were selected from the authors' research to represent a range of non-specialists (e.g., peer mentors, natural counselors, and community health workers), in a variety of settings, providing different interventions (Table 1).

### Data collection and analysis

Authors from this review completed a template regarding their non-specialist training efforts. Materials for the information gathered included their training protocols, field notes from training and supervision, and evaluations they had conducted on these trainings. The template included information regarding who the non-specialists were, the interventions they delivered, their role in treatment, the content of training, training techniques used (e.g., role play and didactic seminars), and lessons learned. The full text of each submitted case study was read by each author to gain familiarity with the content. Two meetings (approximately 1 h long) were held to discuss and refine overarching themes related to the content and techniques used for training and supervision. Additionally, the authors discussed how these themes fit with the broader literature on training and supervision of non-specialists from the literature reviewed. Themes and specific examples from the case studies (Table 1) were then integrated with an overview of literature to inform the considerations for training non-specialist providers in global mental health.

### *Case Study 1: Tuko Pamoja – Family counseling in Kenya*

Case Study 1 took place in Kenya with religious and community leaders conducting a modular family counseling intervention for families with adolescents who had behavioral or emotional concerns (Puffer et al., 2020, 2021). The non-specialists were identified as "natural counselors," with previous volunteer experiences, but did not have prior training in health or mental health provision.

### *Case Study 2: Brief relaxation and trauma healing intervention for Ethiopia*

Case Study 2 was conducted in primary care clinics in Ethiopia with primary care health officers and nurses providing five sessions of treatment for post-traumatic stress disorder (PTSD) to adults with comorbid PTSD and serious mental illness (Ng et al., 2021). In this case study, the non-specialists were employees with past training in the health system and had completed the 2-week mhGAP training prior to being trained in this intervention.

### *Case Study 3: Substance use screening and brief intervention in Kenya*

In Case Study 3, peer mentors (ages 18–26) provided a brief substance use intervention to adolescents living with HIV who screened positive for moderate and high-risk substance use. The intervention included psychoeducation and motivational interviewing techniques to reduce or stop substance use. Peer mentors

**Table 1.** Non-specialist training case studies

| | Case Study 1: Tuko Pamoja: Family counseling in Kenya | Case Study 2: Brief relaxation and trauma healing intervention for Ethiopia | Case Study 3: Substance use screening and brief intervention in Kenya | Case Study 4: Lay health workers enhancing engagement for parents (LEEP) in parent–child interaction therapy in the United States |
|---|---|---|---|---|
| Intervention | Tuko Pamoja ("We are Together" in Kiswahili) Family counseling using systems- and solution-focused family therapy strategies adapted for context and providers; a modular intervention to match content with presenting problems | Brief relaxation and trauma healing intervention for Ethiopia (BREATHE Ethiopia) Treatment for post-traumatic stress disorder symptoms (5 sessions); adapted to the local context and health setting | Substance use screening and brief intervention Brief substance use intervention (1 session); adapted from the World Health Organization ASSIST-Y[a]-linked brief intervention for hazardous and harmful substance use; includes psychoeducation on substance use and motivational interviewing | Parent–child interaction therapy (PCIT) Engagement support (e.g., enrollment, adherence, retention) for Spanish-speaking families in PCIT, an evidence-based parenting program for young children |
| Target population | Families with adolescent children who report problems in relationships and emotional or behavioral concerns | Adults who have PTSD symptoms and comorbid serious mental illness | Adolescents living with HIV who screen positive for moderate and high-risk substance use based on the ASSIST-Y[a] | Spanish-speaking families seeking mental health treatment for their young children with behavior challenges |
| Setting | Kenya (Eldoret) Community-based in family homes | Ethiopia (Sodo and South Sodo District) Primary care clinics | Kenya (Eldoret) Adolescent health clinic in a public, tertiary-level health facility (primarily for youth with HIV) | United States (California) Nonprofit community mental health organization serving low-income families |
| Non-specialist provider | Members of community and religious organizations | Primary care health officers and nurses | Peer-mentors based in the clinic; youth 18–26 years old | Promotoras de salud |
| Previous training and experience | *Previous training*: Majority had no previous mental health training; some had brief trainings in specific topics or approaches (e.g., HIV testing-related counseling) *Experience:* Acting "natural counselors" in the community; nominated by leaders as individuals already sought out by others for advising on sensitive problems (family/parenting issues); most held related volunteer roles (e.g., Sunday School teacher) | *Previous training:* High school-level clinical training in general physical health care; 2-week mhGAP training in mental health diagnosis, psychosocial support, and medication management, primarily for depression and serious mental illness *Experience:* Providing general physical healthcare, medication management, and psychosocial support, primarily in the form of advice giving | *Previous training:* Brief training on general counseling skills, adherence counseling for antiretroviral treatment; reproductive health; and psychosocial support for adolescents *Experience:* Offering adherence counseling and general psychosocial support for issues that youth may have (e.g., conflict with peers and parents) | *Previous training*: General training in the role of promotoras in enhancing community health. Specific training in providing outreach and support for a range of projects related to supporting health and behavioral health interventions *Experience:* Typically serving as a bridge to services (e.g., providing referrals at health fairs); most had experience working with parents in volunteer roles in the past |
| Non-specialist role | The community-based "natural counselors" provide the family counseling, including active facilitation of communication and problem-solving in the session. They use the manualized content and respond flexibly based on the family's responses. They participate in selecting intervention modules and refer participants for more intensive treatment when needed | Health officers/nurses deliver the 5-session treatment for PTSD symptoms that includes psychoeducation about trauma, PTSD, and related symptoms, breathing retraining coaching and practice, and positive coping skills selection and practice. They refer patients experiencing active suicidal ideation to further services as needed | Peer mentors provide the brief substance use intervention to youth with moderate- and high-risk substance use, including providing psychoeducation and using motivational interviewing techniques to build motivation toward reducing or stopping substance use; refers patients to more intensive treatment for those with high-risk substance use | In LEEP, promotoras provide auxiliary support for families that meet criteria for PCIT to enhance engagement, including psychoeducation during enrollment, support to enhance adherence to home practice, and motivation to continue with treatment |
| Training content | 10-day training: Information about family systems and intervention theory of change; training in basic counseling skills applied to families, methods for applying the modular structure and manual, and | 4-day training: Information about PTSD symptoms, intervention theory of change, and research related to the intervention; training on intervention manual and all treatment modules; | 5-day training: Information about substance use symptoms; training in basic counseling skills, motivational interviewing techniques, and treatment | 5-day training: Information about the theory of change and content of PCIT, including how to model and explain targeted parenting skills. and coordination with therapists to promote |

*Cambridge Prisms: Global Mental Health*

**Table 1.** (*Continued*)

| | Case Study 1: Tuko Pamoja: Family counseling in Kenya | Case Study 2: Brief relaxation and trauma healing intervention for Ethiopia | Case Study 3: Substance use screening and brief intervention in Kenya | Case Study 4: Lay health workers enhancing engagement for parents (LEEP) in parent–child interaction therapy in the United States |
|---|---|---|---|---|
| | in delivering the specific manualized treatment modules | training on use of intervention materials and completion of forms and documentation | sessions; training in administering the screening tool | family progress in treatment; provision of iBook with scripts, fidelity checklists, and video supports |
| Training and supervision methods | *Trainers:* Clinical psychologists (US and Kenya) with students (US and Kenya) *Methods:* Didactics; demonstrations; role plays with feedback using fidelity and competency ratings *Supervision:* Sessions were recorded, reviewed by medical psychology students, and rated for fidelity and competency. A tiered supervision model was followed in which medical psychology students met weekly with community-based counselors and reported case summaries and challenges to the clinical psychologists; psychologists gave feedback that students relayed back to counselors; they used role plays to prepare for the following sessions | *Trainers:* American and Ethiopian clinical psychologists *Methods:* Didactics; demonstrations (video clips and live); role plays with feedback *Supervision:* Supervisors were local clinical psychologists who were supervised by the U.S. clinical psychologist. After training, competency was assessed using a behavioral observation. Providers went on to provide care to one patient through a "pre-pilot" during which they received bi-weekly in-person supervision and in-person observation. Providers who met clinical competency on the ENhancing Assessment of Common Therapeutic factors (ENACT) tool and fidelity to the intervention using BREATHE Ethiopia specific ratings scales were then assigned a case load and received weekly phone supervision. Sessions were audio recorded and rated for fidelity to inform supervision | *Trainers:* Local Kenyan psychiatrists and psychologists *Methods:* Didactics; quizzes; role plays with feedback using fidelity and competency ratings *Supervision:* Weekly supervision was provided by local psychologists and psychiatrists. Sessions were audio recorded and rated using fidelity checklists to inform supervision. | *Trainers:* Clinical psychologist with students *Methods:* Didactic presentations; live demonstrations; role plays with feedback using behavioral observation of parenting and promotoras skills *Supervision:* Bi-weekly supervision was provided by clinical psychology doctoral students overseen by a licensed psychologist, supervision focused on case review, care coordination, and technical assistance. Sessions were video recorded and reviewed to inform supervision |
| Primary publications | Puffer et al. (2020, 2021) | Ng et al. (2021) | Jaguga et al. (2022) Original intervention: Humeniuk et al. (2010) | Barnett et al. (2019), Davis et al. (2022) |

[a]ASSIST-Y, Alcohol Smoking and Substance Involvement Screening Test – Youth version.

also made referrals to more intensive treatment for patients with high-risk substance use.

### Case Study 4: Lay health workers enhancing engagement for parents

The fourth case study is from the United States, with promotoras de salud supporting engagement for Spanish-speaking immigrant parents in an evidence-based parenting intervention, parent–child interaction therapy, provided within a community mental health setting (Barnett et al., 2018a,b; Barnett et al., 2019; Davis et al., 2022). In this intervention, non-specialists were trained to promote enrollment, adherence, and retention in an intervention that was provided by professional mental health providers.

## Considerations for training non-specialists in evidence-based psychological interventions: In integration of research literature and case studies

### Who are the non-specialists?

When developing training for non-specialist providers, it is important to consider what their previous training and experiences are and how they normally interact and work with the population of interest. This can help identify existing strengths that may be leveraged, and gaps that will need to be addressed within training. The mhGAP refers to primary health care staff (e.g., nurses) and lay health workers (e.g., peer providers and community health workers) as non-specialist providers involved in task-sharing models. All of these non-specialists have varied professional and personal backgrounds which impact the training they need to receive. For example, primary health care staff being trained to deliver a mental health intervention already have training related to healthcare systems (e.g., documentation and ethics of patient confidentiality), whereas lay health workers may not have this background and require training beyond the specific intervention. On the other hand, lay health workers often have shared lived experiences, including having similar clinical diagnoses or family experiences (e.g., peer providers and parent partners) or being from the same community with similar cultural backgrounds (e.g., community health workers and promotoras de salud) as the populations being served (Gustafson et al., 2018; Jack et al., 2020). These experiences can increase their expertise in strategies on how to engage individuals in care, including building trust through their shared identities and experiences (Gustafson et al., 2018; Barnett et al., 2021). However, the literature suggests that these shared experiences can also make it challenging for peer providers to maintain boundaries, and therefore it is important to address how the providers balance their past experiences with their new responsibilities (Satinsky et al., 2021).

The case studies provided in this article demonstrate how critical it is to take into account the type of non-specialist you are working with and their previous experiences when developing training in psychosocial and psychological interventions. In the case studies, all non-specialists had various levels of previous training and experiences related to physical and mental health services. Whereas the religious and community leaders in Case Study 1 providing Tuko Pumoja to families had previous volunteer experiences and were seen as "natural counselors," the primary health care officers and nurses in Case Study 2 providing BREATHE to adults with co-morbid PTSD and severe mental illness had a background in general physical health care and had already completed a 2-week mhGAP training. These varied backgrounds impacted the content of training that was covered across these two programs, such as different emphases on basic counseling skills. Therefore, when designing a training for non-specialist providers, it is important to consider what the baseline skillset is of the trainees, as certain content may need an increased focus for providers without any past training in mental health.

### What is the non-specialist role in intervention delivery?

The next key question to ask when planning training for non-specialist providers is the role that they will be taking in mental health care provision. Non-specialists can increase access to mental health services with the following roles: 1) screening and navigation to care, 2) auxiliary support (e.g., case management, motivational enhancement), 3) stepped-care models, and 4) primary providers of interventions (Barnett et al., 2018b). These roles vary in how involved professional providers are in care. With navigation and auxiliary support, professionals remain the primary treatment providers, with non-specialists providing psychosocial interventions that promote enrollment and engagement in care. Task-sharing roles include stepped-care models where non-specialists provide prevention-level services, with the expertise of professional providers saved for more clinically intensive cases (Patel et al., 2008). Finally, non-specialists are frequently the primary providers of all services in settings with few mental health professionals (Barnett et al., 2018a; Bunn et al., 2021). Often the roles and responsibilities of the non-specialists are determined by the structure of health service delivery and financing, the availability of specialist providers, and the needs and preferences of consumers, in each specific context. These case studies described reflect a common difference between roles in LMICs, which usually have very limited mental health specialist availability, and HICs, where non-specialists may have auxiliary roles to professional providers. In our case studies, the non-specialists in Case Studies 1 and 2 were the primary providers of the mental health interventions, whereas the promotoras de salud in Case Study 4 supported treatment engagement in parent–child interaction therapy, which was provided by professional providers in the United States. Case Study 3 included a stepped care model with peer mentors providing psychoeducation and motivational interviewing for one session and referrals to professionals for more intensive treatment needs.

Along with identifying the role that the non-specialist is going to have in the mental health service delivery, it is important to understand how their current roles may interfere with learning and implementing an intervention. For example, non-specialists in primary care clinics may not have adequate time to deliver psychosocial or psychological interventions with their competing job demands (Baker-Henningham et al., 2005), and peer providers might see delivering structured treatments to be outside the scope of their practice (Magidson et al., 2019). Poor role definition, increased work pressure, and challenging relationships with specialized mental health professionals have all been identified as barriers to task-sharing mental health interventions, and therefore should be addressed as part of an implementation effort (Bunn et al., 2021).

### Which skills are needed?

Psychosocial and psychological interventions require skill-building in common factors (e.g., rapport building and demonstrating empathy) and intervention-specific components (Pedersen et al., 2020). Intervention components could include

cognitive restructuring in cognitive behavioral interventions, exposure for anxiety, or teaching parents how to reinforce positive behaviors in their children. When training professional mental health providers in an evidence-based psychological intervention there is an assumption that clinicians have basic knowledge and skills in common psychotherapeutic factors (e.g., nonjudgmental attitude, reflective listening, welcoming nonverbal communication), however, this may not be the case for non-specialists. Whereas some have advocated that certain non-specialist providers have a natural skill set for building trust with communities they belong to (Gustafson et al., 2018), others have highlighted a need to evaluate these skills (Anvari et al., 2022). Additionally, psychoeducation about mental health symptoms and disorders, and understanding how to conduct case conceptualization and how to apply the underlying theoretical bases of interventions (e.g., behavioral principles) facilitates the successful implementation of interventions (Murray et al., 2011; Atif et al., 2019). Including the rationale for the intervention is important so that non-specialists better understand how to tailor the intervention to the individuals they see (Murray et al., 2019).

Clearly, the goals of training non-specialists vary based on their level of involvement in delivering the mental health intervention. For example, both Case Study 1 (providing community-based, family counseling in Kenya) and Case Study 4 (engagement support for parent–child interaction therapy in the United States) trained non-specialists to work with families to address challenging behavior in children. However, in Case Study 1, non-specialists were the primary providers of the intervention, whereas in Case Study 4, they provided auxiliary support (e.g., psychoeducation about the intervention, motivation enhancement) to families receiving care from a professional provider. Therefore, the training content had varying levels of emphasis on the ability to deliver the intervention, versus being able to explain the intervention and support adherence with skills modeling (see Table 1).

### How are the skills taught and evaluated?

The previous three questions related to the non-specialists, their roles, and the skills they need to inform the development of training and supervision models. The final question relates to the techniques that best support skill uptake and continued use. In general, training methods have been underreported on studies with non-specialists, which limits our understanding about what techniques and training intensity are most effective for non-specialists (Barnett et al., 2018a; Caulfield et al., 2019; Bunn et al., 2021). Initial training programs have been described as varying greatly in duration from a few days to multiple months (Fayyad et al., 2010; Patel et al., 2010; Magaña et al., 2015). Regarding how the skills are taught, training activities usually include didactic presentations, discussions, with emphasis on active learning strategies (e.g., role playing) to practice new skills and receive immediate feedback (Barnett et al., 2018b; Bunn et al., 2021). Active learning strategies are seen as the most effective ways to train providers, including professionals, in psychological interventions in general (Valenstein-Mah et al., 2020), and likely is even more critical with non-specialists with less familiarity with these skills. Notably, in all four case studies, role plays with feedback were seen as a critical learning strategy in initial training and ongoing supervision. This helped to reinforce and monitor competencies in intervention delivery.

It has been recognized in the literature and the experiences outlined in our case studies that ongoing training in the form of consultation or supervision is needed to support case conceptualization, skill maintenance, and the delivery of interventions with fidelity (Frank et al., 2020). Notably, the need for ongoing supervision and consultation has been recognized as critical for any implementation effort, including for mental health professionals (Schoenwald et al., 2004; Edmunds et al., 2013), and it has not been established whether non-specialists require different levels of ongoing training (Barnett et al., 2018a). Though the majority of global implementation efforts have had supervision and consultation provided by mental health professionals from HICs, increasing efforts have local non-specialists serve in these roles (Murray et al., 2011; Dorsey et al., 2020a,b). Indeed, in our case studies, local supervisors supported ongoing training with non-specialists for the interventions conducted in Kenya and Ethiopia. These efforts hold possible advantages for enhancing the sustainability of psychological interventions in global settings. Supervision or ongoing consultation is especially important to address complex clinical issues, such as risk of harm to self and others. Furthermore, it is important to address the impact that intervention provision might have on non-specialists, many who also have experiences of trauma, poverty, and discrimination, to prevent burnout and vicarious traumatization (Jain, 2010).

Ongoing monitoring has been recognized as an important component of making sure that non-specialists are delivering psychosocial and psychological interventions with content fidelity and competency (Kohrt et al., 2018; Anvari et al., 2022). Content fidelity refers to the degree to which the provider delivers an intervention in the way it was intended, whereas competency refers broadly to knowledge and skills the provider has in delivering an intervention and often includes more non-specific factors (Gearing et al., 2011). For example, the ENhancing Assessment of Common Therapeutic Factors (ENACT) tool was developed to measure non-specialist competence (Kohrt et al., 2015), and was adapted for use within Case Studies 1 and 2 along with an intervention-specific fidelity monitoring to evaluate how the non-specialists were delivering the BREATHE and Tuko Pumajo interventions. Ongoing monitoring of competence and intervention-specific fidelity allowed for an ability to focus supervision to the specific needs of providers.

It is important to approach fidelity measurement with respect for the unique characteristics and skills that non-specialists bring to intervention delivery. For example, recent research measured both the fidelity of peer recovery specialists in delivering the content of an evidence-based intervention for substance use, and the use of appropriate self-disclosure (Anvari et al., 2022). Additionally, it has been identified in trials in Kenya and India that fidelity measures might miss the adaptations that non-specialists make to interventions, which could increase engagement or could be potentially harmful to the intervention (Leocata et al., 2021). Strategies to capture these adaptations could enhance our ability to improve interventions with non-specialist cultural expertise, while training non-specialists around the types of adaptations that could be inappropriate.

An important area of increasing focus in training non-specialists has been on the role of technology (e.g., mobile apps and online platforms) in promoting scalability of training non-specialists (Naslund et al., 2019; Rahman et al., 2019; Triplett et al., 2021; Nirisha et al., 2023). For example, Case Study 4 leveraged technology by providing the promotoras de salud with an e-book embedded with intervention scripts and videos to help maintain fidelity and demonstrate skills with families. Technology can be incorporated in task sharing in various ways to enhance the non-specialists' ability to connect with their clients and/or trainer (Naslund et al., 2019; Triplett et al., 2021). Furthermore, digital training could expand the number of non-specialists who can be trained. A

comparison of specialist delivered face-to-face training and technology-assisted in delivering an evidence-based intervention for perinatal depression found similar levels of competence across training conditions, with the technology-assisted training costing 30% less (Rahman et al., 2019). Future research is warranted to identify how digital technologies increase the number of non-specialists trained, the quality of intervention delivery, and clinical outcomes for clients.

## Conclusions

Scientific literature and our case studies highlight important themes to training and supporting non-specialists to provide high-quality delivery or support of psychosocial and psychological interventions. First, there is not a one size fits all approach to these training efforts, as the context of the non-specialists impacts implementation including, their prior training, experiences and previous roles, cultural expertise, and organizational support. Therefore, it is important to develop implementation strategies to consider the provider context in addition to the intervention itself; developing training and support strategies in collaboration with non-specialists is likely to enhance fit. Second, training, consultation, supervision, and ongoing fidelity monitoring should focus on content that extends beyond a single intervention and includes overall competency with common factors (e.g., rapport building) and understanding of the rationale for interventions. This prepares non-specialist providers to make informed clinical decisions and to respond flexibly based on presenting challenges. This also allows providers to blend their cultural expertise with their understanding of the intervention to help adapt it for the communities they serve. These adaptations have the potential to (a) enhance engagement in the interventions, which could improve implementation and sustainment over time and (b) to improve the relevance and understandability of the content itself in ways that could improve clinical efficacy. Third, similar to professional mental health providers, the methods used to train and supervise should include ample opportunities for role play and session review to allow for personalized feedback, as these methods are critical to enhance competency. For all of the above points, there remain empirical questions for future research to identify the key ingredients for effective training and support across types of providers and interventions.

Overall, expanded use of these recommended practices has the potential to move the field of training non-specialist mental health providers beyond the traditional approach of focusing solely on the intervention under investigation, to strategies that can truly build a workforce capable of addressing the global mental health gap. At the same time, it is critical to recognize that training and supervision are only part of the implementation strategies, or techniques to facilitate scalable and sustainable delivery of psychosocial interventions (Powell et al., 2015). A recent review found that implementation of task sharing is often impacted by structural challenges, including societal stigma around mental health, limited financing for mental health care, and challenges integrating non-specialists into systems of care. The authors noted that, "the science of developing implementation strategies that could be used to address barriers or to leverage facilitators of task-sharing mental health interventions across all levels is only in its nascent stages" (Le et al., 2022, p. 20). Future research is needed to identify and test implementation strategies, beyond training and supervision, associated with high-quality task sharing.

In sum, we advocate for a training approach in global mental health in which non-specialists are recognized as being specialists in serving their communities – and are further empowered in being partners in developing solutions to address mental health gaps. To make this a reality, we need to identify feasible pathways for non-specialists to continue to support how this workforce grows and contribute to scale-up efforts. For example, non-specialists who gain expertise in interventions can gain employment as the local trainers and supervisors (Dorsey et al., 2020a,b). Current non-specialists are among the most qualified individuals to expand mental health workforce capacity, and global mental health researchers and practitioners should partner with them to pave the way.

**Open peer review.** To view the open peer review materials for this article, please visit http://doi.org/10.1017/gmh.2023.19.

**Acknowledgments.** The authors would like to acknowledge the non-specialist providers that they have partnered with to understand how to best train and support this important workforce. Additionally, the authors would like to thank Hanan Salem for her support in manuscript organization.

**Financial support.** This research and effort was supported by the National Institute of Health K01MH110608 awarded to M.L.B.; K23MH110601–04 awarded to L.N.; the Fogarty International Center of the National Institutes of Health (D43TW009345) awarded to F.J. through the Northern Pacific Global Health Fellows Program; and Grand Challenges Canada and Duke Global Health Institute awarded to E.P.

**Competing interest.** The authors declare none.

**Ethics standard.** The authors assert that all procedures contributing to this work comply with the ethical standards of the relevant national and institutional committees on human experimentation and with the Helsinki Declaration of 1975, as revised in 2008.

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
