## [Reviewer Report]

We are pleased to submit the invited manuscript, Effective Training Practices for Non-Specialist Providers to Promote High Quality

Mental Health Intervention Delivery, as part of the relaunch of the Cambridge Prisms: Global Mental Health journal. The work presented has not been submitted to any other journals. We look forward to your review.

---

## [Reviewer Report]

*Comments to Author*: The authors’ rationale for the review is couched within the context of needing to address the global health knowledge gap in relation to implementation strategies that would enhance implementation and scale up of task shared evidence-based practices (which would have been a great review!). 

However, they then indicate that the purpose of the review is to provide a conceptual review of the state of the science related to the training and ongoing support (e.g., training and supervision) needed for non-specialist providers to deliver or support evidence-based practices with fidelity. The first concern is thus a mismatch of their stated objective with the rationale for the review which is based on the need for evidence on implementation strategies that enhance implementation and scale up of task shared evidence-based practices. 

The second major concern, and more alarming, is that it is unclear how this “conceptual” review was undertaken as there is no methods section. It appears that what they did to achieve their stated objective was to present and discuss four examples of non-specialist training in relation to the literature. How the case studies were selected (none describing any attempts at broader scale-up) and how they gleaned the information from the case studies is unclear – there being no methods section. Given the lack of any scientific approach used for the “conceptual review”, as well as just a few arbitrary case studies being used, the veracity of the conclusions drawn are thus questionable.

---

## [Reviewer Report]

*Comments to Author*: While a useful summary of some of the approaches approaches to training of non-specialists in mental health, the article misses some key areas of development in the field. Especially conspicuous by its absence is a discussion of the use of technology to assist with training, supervision and their scale-up. (see for e.g., Naslund et al Harv Review Psychiatry 2019; Rahman et al Global Mental Health 2019; Nirisha et al Community MH J 2022). This is a key area for future development globally and the article should include this recent research in its review.

---

## [Reviewer Report]

*Comments to Author*: This manuscript provides a description of training-related implementation strategies to support non-specialist providers in delivering mental health care in global settings, using case examples to illustrate these strategies. Overall, this manuscript was very well written and covered an important topic that is relevant for expanding global access to mental health interventions. The authors made a compelling argument for why to include studies from HIC and LMICs. I have a few suggestions to improve the manuscript: 

1. The introduction provides a strong case for why non-specialists are well suited to address the global mental health gap. To balance this, it might be helpful to include a brief discussion somewhere in the manuscript about the potential big picture challenges of the task sharing approach (e.g., implications for professional providers if there is a larger non-specialist workforce, sustainability of training initiatives and reimbursement for the non-specialist workforce). 

2. I would suggest slightly reframing the scope of this study. Given that there is not a clear discussion of methodology (or a methods section), it seems inaccurate to refer to it as a review. It seems more like a series of case examples, a brief overview of the evidence for implementing mental health interventions by non-specialist providers, and suggestions for training-related implementation strategies to support non-specialist providers. Given that the focus was primarily on training-related implementation strategies and that other implementation strategies were given less attention, it might be helpful to set up that expectation in the introduction. Alternatively, if the authors want to highlight non-training related implementation strategies, perhaps these could be summarized in a table or in an additional section of the manuscript. 

3. I recognize that there are space constraints, but it would be helpful to have a little bit more description about each case example in text so that the reader does not have to rely solely on the table for this key aspect of the manuscript. A minor point, but there are specific references to Case Example #1 (and #2) in text, but this does not correspond to numbering in the table or a description of the case examples in text. 

4. There were a few minor typographical errors:

- Headings: Except for the “Conclusions” section, all headings are subheadings of the “introduction” section. It seems like there should be other headings that are centered / not sub-headings prior to the conclusions section. 

- Page 5: In the section, “who are non-specialists?” – the fourth line in that section says “within the training, consultation, and supervision”

- Page 7: The sentence at the end of the first paragraph is very long and challenging to follow

- Page 9: typo in the first sentence – seems to be missing the word “to” before “inform”

- Page 10: the word “enhance” is used twice in the same sentence

---

## [Reviewer Report]

*Comments to Author*: The information presented in this paper so far stands as a starting point for a fuller and more complete paper that can successfully carry out the full goals articulated in the abstract and the paper itself. The overarching problem is that it is unclear as to whether this is a conceptual review of the state of the science (if so there is too little information and the methods are unclear), or is this a presentation of case studies completed by the collective of Authors (if so this is not states clearly either and the methods are unclear

The paper states that:

“The current review provides an overview of the evidence concerning the training and ongoing support of non-specialists carrying out task-sharing mental health care models while utilizing case examples from studies that took place in Kenya, Ethiopia, and the United States.”

This is a broad goal. Is it an overview of evidence concerning training, or is it a paper presenting case studies? It primarily presents the experience of a limited number of case studies. The paper does not provide a comprehensive overview of this topic, and a review of the evidence cannot happen from only the cases mentioned. “Training” and “ongoing” support” are separate areas of focus. Training content and training supervision methods are presented as the main components of “support”. Successful implementation of community-based programs for mental health care delivery involves much more than training. Ongoing support can include financial enablers, calculating task time based on volunteer employee status, assignment of tasks, recruitment of the right people based on the task sharing strategy, how to train for success, and how to supervise and mentor for excellence and fidelity. There are many issues that go into comprehensive support of non-specialist providers. The paper is unclear about training for what, either: delivery of psychosocial interventions; delivery of psychological interventions; case finding; psychiatric rehabilitation for more severe conditions; basic social support; or other. It seems the paper is focused on non-specialist providers of psychological interventions, which may be the main research finding of the field of global mental health, but this is not stated clearly from the outset. The stated goals of the paper are therefore difficult to ascertain, based on the content of the paper. The literature cited in the case studies is mostly the authors’ own. If this paper is focused only on implementation of training programs, it should be clear to only focus on training, and not imply that the paper addresses broader issues of program implementation of non-specialist providers. If this paper is a summary of evidence the scope of review needs to be broader than the included references. A paper with the stated goal above would need to cover a broader set of topics, and review evidence beyond a number of case studies with attached publications. 

The paper states that:

“In this review, the authors discuss the various implementations strategies that can be used when training and supporting non-specialist providers and highlight the importance of tailoring these strategies to the skills and needs of the non-specialist providers and their roles in the delivery of an intervention model.”

The implementation strategies involved around the training of non-specialist providers, and the actual implementation of their work, is more involved than what is discussed in this paper. What stands in the paper is a basic review of implementation strategies for training and supporting non-specialist providers, and it could be fuller. The issue of “supporting” is a significant one, deserving of a more developed review, as above. 

The paper states that:

“This review advocates for training and implementation strategies that provide non-specialist providers with opportunities to develop into specialists, promote culturally responsive care within their communities, and expand the mental health workforce.”

This seems to defeat the overarching purpose of task-sharing in mental health care delivery. Who will replace them if they become specialists? This seems to be a call for systems of professional development to be established that can create a stream of engagement across different levels of task-shared systems. Attention to implementation of such systems is not included in the paper. A systems of care approach is lacking in the paper. 

The paper states that:

“This conceptual review will discuss the state of the science related to the training and ongoing support (e.g., consultation, supervision) needed for non-specialist providers to deliver or support EBPs with fidelity to address the global mental health gap.”

The paper does not do a thorough job of discussing the state of the science. It is unclear also what EBPs are being discussed. EBPs can include preventive interventions, social support, psychosocial interventions, psychological interventions, pharmacologic interventions, etc. A search of the term “psychological” comes up many times in the References, but not once in the text of the actual paper. The core assumptions underlying the paper itself on the scope of work it encompasses are unclear. It would be helpful if the authors worked to clarify these points.

One reviewer notes that there are missing key elements in the field of training of non-specialists in mental health, including technology. Another reviewer notes that there is no scientific approach and two reviewers note no Methods section to explain the organizing principles and methods underlying the Authors’ approach to the paper. The quality of the writing can be improved. 

There is merit in the topic and also in the aims of the paper, depending on which one the Authors choose to deepen. A major revision would be needed for publication, to address the concerns of the reviewers and major clarifications here recommended. If the authors need to exceed wordcount please state that this will be needed to ensure all comments are addressed. 

Thank you and best regards.

---

## [Reviewer Report]

Dear Editorial Board,

We appreciate the opportunity to submit a revised manuscript for the invited paper, “Effective Training Practices for Non-Specialist Providers to Promote High Quality Mental Health Intervention Delivery.” The manuscript was strengthened by reviewer and editor comments and we are hopeful it makes a strong contribution to the relaunch of Cambridge Prisms: Global Mental Health.

Sincerely,

Miya Barnett

---

## [Reviewer Report]

*Comments to Author*: Unfortunately, I do not believe that that the authors have adequately addressed my concerns. While they have indicated that they have reframed the article to focus on training rather than implementation strategies for task sharing of psychological and psychosocial counselling (and notwithstanding that they still refer to implementation strategies in the manuscript), they have not broadened the scope of the review beyond the four case studies. Their justification for these four case studies being that they “represent a range of non-specialists (e.g., peer mentors, natural counselors, community health workers), in a variety of settings, providing different interventions” is insufficient. Even if they provided more detail in this regard, four cases are not sufficient for a review and they certainly cannot talk about effective training strategies based on these four case studies. Further, rather than providing an adequate methodology section which is important to be able to assess the scientific value of the findings, they have chosen to indicate that they are not obliged to provide a methodology section. Instead, they have chosen to just indicate that they used a common framework for evaluating the four case studies which is in adequate.

---

## [Reviewer Report]

*Comments to Author*: The authors were highly responsive to reviews and the revised manuscript is improved as a result. I especially appreciate the clarification about the scope of the manuscript, as well as the inclusion of brief descriptions of each of the case examples in text. In my re-reading of the manuscript, I noticed the following minor typos: 

Pg. 4: “behavior activation” should be “behavioral activation”

Pg. 9: “whereas, some” – remove comma 

Pg. 13: “and or trainer” should probably be “and/or trainer”

Pgs. 14-15: the phrase “at the same time” is used twice in the same paragraph

---

## [Reviewer Report]

*Comments to Author*: By revising the scope of the paper and addressing some key gaps such as the use of technology in training, I feel the authors have done a satisfactory job and the paper is acceptable for publishing.

---

## [Reviewer Report]

*Comments to Author*: The authors have responded well to the reviewer comments and the detailed responses are greatly appreciated. The actual review of the literature and evidence comes to a total of about seven or eight paragraphs in which the case examples are not being discussed, which seems to qualify it as brief. There remains a bit more work to clarify at the outset what the scope of the paper is. This can help to address clearly stated concerns about the language around it being a review, and the lack of a methodology section. Highlighting the use of case examples in the title will also help. Care should be taken to be clear and consistent throughout the paper 

I suggest at minimum:

-In the title add: “…: Four Case Examples from Kenya, Ethiopia and the United States”, that is, the title would be: Effective Training Practices….Delivery: Four Case Examples from Kenya, Ethiopia and the United States

-In the abstract change “an overview” to “a brief conceptual overview”

-In the abstract replace “In this review” with “In this paper”

-In Impact Statement change “Case examples and a review of the literature…” with “Case examples and a brief narrative review of the literature…”

One more detailed read-through of the paper by the authors to improve and optimize readability, consistent with some of the reviewer feedback, would be appreciated. 

This is valuable and important work represented in the paper, and will contribute to our understanding of existing and best practices. Thank you.

---

## [Reviewer Report]

Dear Editors,

Thank you for the correspondence to support the revision of this invited review paper. We believe we have been responsive to the Editors and Reviewers, and hope that the current version fits with the journal. 

Sincerely,

Miya Barnett